# Explicable Policy Search

**Ze Gong**
Arizona State University
Tempe, AZ 85281
zgong11@asu.edu

**Yu Zhang**
Arizona State University
Tempe, AZ 85281
yzhan442@asu.edu

## Abstract

Human teammates often form conscious and subconscious expectations of each other during interaction. Teaming success is contingent on whether such expectations can be met. Similarly, for an intelligent agent to operate beside a human, it must consider the human's expectation of its behavior. Disregarding such expectations can lead to the loss of trust and degraded team performance. A key challenge here is that the human's expectation may not align with the agent's optimal behavior, e.g., due to the human's partial or inaccurate understanding of the task domain. Prior work on *explicable planning* described the ability of agents to respect their human teammate's expectations by trading off task performance for more expected or "*explicable*" behaviors. In this paper, we introduce *Explicable Policy Search* (EPS) to significantly extend such an ability to stochastic domains in a reinforcement learning (RL) setting with continuous state and action spaces. Furthermore, in contrast to the traditional RL methods, EPS must at the same time infer the human's hidden expectations. Such inferences require information about the human's belief about the domain dynamics and her reward model but directly querying them is impractical. We demonstrate that such information can be necessarily and sufficiently encoded by a surrogate reward function for EPS, which can be learned based on the human's feedback on the agent's behavior. The surrogate reward function is then used to reshape the agent's reward function, which is shown to be equivalent to searching for an explicable policy. We evaluate EPS in a set of navigation domains with synthetic human models and in an autonomous driving domain with a user study. The results suggest that our method can generate explicable behaviors that reconcile task performance with human expectations intelligently and has real-world relevance in human-agent teaming domains.

## 1 Introduction

Intelligent agents are quickly becoming a part of our daily lives in a variety of domains, including smart home, autonomous driving, education, and so on. In such domains, the agents are expected to perform in human inhabited environments and even collaborate closely with them. Members in such a collaborative setting often form conscious and subconscious expectations of each other and the success of collaboration depends on whether such expectations can be met. A key challenge here is that the human's expectation may not align with the agent's optimal behavior. Hence, agents choosing their optimal behaviors without considering the human observers or collaborators could be seen as unexpected, leading to degraded team performance and loss of trust [19, 6].

Consider a drone navigation scenario in Fig. 1a where a drone is tasked to navigate from the starting position (as shown) to a destination. On a good day, the drone would be expected to navigate (almost) straight to the goal under mild air perturbation (thus possibly introducing a slightly curved trajectory as illustrated in red). When there is a heavy wind that changes the domain dynamics (which the human observer is unaware of), it may become impossible for the drone to navigate as usual. Two

36th Conference on Neural Information Processing Systems (NeurIPS 2022).

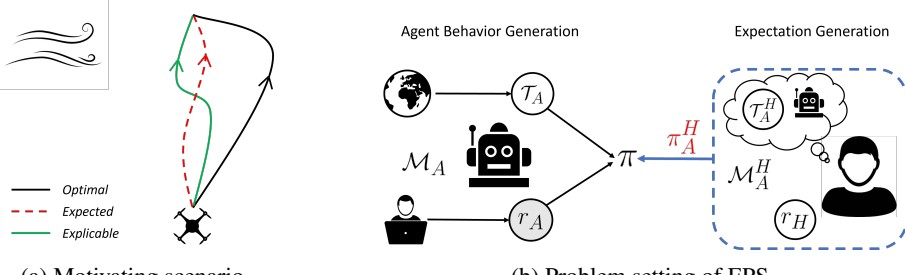

(a) Motivating scenario.

(b) Problem setting of EPS.

Figure 1: The agent learns its behavior with a given engineered reward model, $r_A$, in the task environment under the true domain dynamics $\mathcal{T}_A$. The human generates her expectation of the agent's behavior $\pi_A^H$ based on $\mathcal{T}_A^H$ that captures her belief or understanding of the true domain dynamics, and her reward model $r_H$, which may differ from $\mathcal{T}_A$ and $r_A$, respectively. Such differences may be due to personal preferences in $r_H$, biases and misunderstandings about the domain dynamics, etc. In the traditional RL setting, the agent computes the optimal policy $\pi$ under its models, which could be different from its expectation formed by the human based on her models. Shaded nodes are known to the learning agent and unshaded nodes are unknown.

alternatives are illustrated where the drone navigates in a zigzag (green) and a curved pathway (black), respectively. The curved pathway is more optimal since it makes fewer sharp turns but the former is likely to be perceived as more expected to the human observer, given its *similarity* to the expected behavior. We note that searching for the explicable behavior goes beyond choosing from or combining the agent's optimal behavior and the human's expected behavior. Hence, achieving such a capability requires a more fundamental treatment. In the experiments, we show that our method can generate novel behaviors that are unseen in the training scenarios (where human feedback is solicited).

The problem setting of *explicable policy search (EPS)* is illustrated in Fig. 1b, which is fundamentally a model reconciliation problem [7]. In such a setting, considering the learning problem only from the perspective of the agent could sabotage the teaming performance when the agent's behavior differs from human's expectation, while considering only from the human's perspective may be undesirable either. Our method considers any feedback the human may provide to the agent based on her expectation of its behavior to help the agent improve its behavior. Hence, our method has a variety of applications where machines must be tuned to individual users, such as an autonomous vehicle that learns to improve its driving behavior based on its owner's feedback but at the same time must abide by standardized operating requirements to ensure safety, comfort, etc. In our problem, we assume that the human is nosily rational at generating her expectation of the agent to accommodate her computational limitations. Noisy rationality is one type of bounded rationality [45] due to a limited computational capability, resulting in sub-optimal decisions. While noisy rationality may not perfectly describe the human's computational model [43] due to various cognitive biases, modeling humans as such has been a common practice and a reasonable assumption in prior work [34, 2]. Furthermore, similar to prior work on explicable planning [48, 25], we assume that the human is a sole observer of the agent. Extending it to an collaborative setting requires additional machinery and will be discussed in future work, with steps already taken in a classical planning setting [47].

In this paper, we formulate *explicable policy search (EPS)* in a model reconciliation setting. The goal is to learn a policy that reconciles maximizing the long-term return (based on the engineered reward model $r_A$) with maximizing the expectedness of the agent's behavior from the observer's perspective, which is quantified by an *explicability* score. We formulate the problem as a simple linear combination of two objectives, which already yields substantial benefits compared to several baseline methods in our evaluation. The challenge in our problem formulation mainly lies in the fact that the human's expectation is hidden (but partially observable via human feedback) and must be learned. A straightforward solution involves learning both the human's belief about the domain dynamics ($\mathcal{T}_A^H$) and her reward model ($r_H$) from human feedback, which is possible but impractical. Instead, we show that the information needed from them for EPS can be necessarily and sufficiently encoded by a surrogate reward function, which is much easier to learn. Such a reward function can then be incorporated into a policy search process to recover an explicable policy. The final learning objective of EPS after the modification bears some similarity to maximum entropy reinforcement learning [20, 21] but is derived under a completely different motivation.

For evaluation, we design a set of navigation domains where the human's feedback on her expectation of the agent's behavior is synthetically generated based on the true domain dynamics modified with various "misunderstandings". We compare our method with three baselines that are selected to represent the traditional RL methods to demonstrate the key advantages of EPS under model differences. Furthermore, to show its real-world relevance, we conduct a human study in an autonomous driving domain [29] where we design driving scenarios to elicit existing human biases about the vehicle's domain dynamics that could be dangerous to ignore during learning. The results show that our method can intelligently generate explicable behaviors that are safe and preferred over those of the baselines.

The contribution of this paper is three-fold. First, we introduce and formulate explicable policy search (EPS), which significantly extends explicable planning to stochastic domains in a reinforcement learning setting with continuous state and action spaces. Second, we propose a practical solution for this challenging problem by introducing a surrogate reward function learned from human feedback data, which encodes the necessary and sufficient information for explicable policy search. Third, we evaluate our learning method extensively with simulations and human subjects.

## 2   Related Work

The problem of generating communicative actions or behaviors (a.k.a. explainable behaviors) has been well studied as a subarea of explainable AI. Explainable behavior generation has a subtle but important difference from interpretable behavior generation [44, 35, 38, 39]. While the former aims at generating behaviors that are inherently and implicitly interpretable to humans, the latter is focused on generating behaviors under interpretable model representations so that their rationales can be explicitly conveyed. In this regard, a behavior generated by the latter may not be interpretable by itself but an explainable behavior will be. In domains where explicit communication should be used sparingly, such as in human-agent teaming, or high-stake domains where "surprises" should be minimized, explainable behaviors would be more desirable. Various terms have been introduced for different aspects of explainable behavior generation such as legibility, explicability, transparency, etc. For a review of their similarities and differences, refer to [7]. A closely related term to our work, legibility, for legible motion planning [12], is about generating trajectories that better reveal the underlying goal of the agent. Explicable planning [48, 25, 16, 47], which is the focus of our work, differs from legible motion planning in that it aims at generating plans that better align with the human's expectation *given the goal*. The key characterization of explicable planning methods revolves around the idea of model reconciliation where an agent makes decisions based on two different models instead of one, with a focus on the differences in domain dynamics models [8, 6, 7]. While there exists prior work [33, 5] on training value-aligned learning agents that learn to respect different reward models, they do not consider the differences in the domain dynamics. Prior methods on explicable planning addressed the problem in a classical planning setting (e.g., PDDL [15]) under deterministic domains. It remains a challenge to extend the problem formulation to stochastic domains with continuous state and action spaces. Also, differing from the prior work on explicable planning, we consider that the user's reward model may also differ from the agent's, but assume the differences do not introduce a discrepancy in the perception of the agent's "goal" (which the user knows). Considering multiple candidate goals is the problem setting of legible motion planning [12].

As shown in Fig. 1b, estimating the human's expectation requires learning both the human's belief about the domain dynamics and her reward model based on human feedback. Existing work on inverse reinforcement learning (IRL) [1, 36, 49] and reward learning [11, 41, 13] can learn the reward model from human data with the assumption that the human has access to an accurate model of the domain dynamics. It was shown that the human's inaccurate belief about the domain dynamics may skew the human's feedback and lead to learning the opposite preferences with respect to the human's true reward model [17]. A similar issue may occur in policy learning for various methods that are devised to use human feedback, which include reward shaping, policy shaping [18], and interactive RL [24, 32, 10, 27]. When the human has an inaccurate belief about the domain dynamics, as we will show, it can lead to non-convergence during learning or high variances in the learned behaviors. While it may be possible to learn the human's belief about the domain dynamics separately [37], the belief and human's reward model are generally tightly coupled in the human's feedback and should be jointly learned. Joint learning is possible [23, 17] but challenging due to its high dimensionality, which makes it impractical for real-world domains. In a similar vein, InfoGAIL [30] uses latent code to discover salient semantic features to accommodate demonstrations from different experts. Even

though the aim there is quite different from ours, one may consider the differences in the domain dynamics as captured by the latent code. However, their work is focused on the "discovery" while we focus on "reconciling" the differences (without explicit modeling them). When considering only differences in the domain dynamics, our problem may be viewed as a form of off-dynamics learning [14] that addresses transfer learning from a source to a target domain. However, we do not have direct access to the target domain in EPS, which is hidden in the human's mind.

## 3 Problem Formulation

In this paper, we formulate any task domain as a Markov Decision Process (MDP). An MDP is represented by a tuple $\mathcal{M} = (\mathcal{S}, \mathcal{A}, \mathcal{T}, r, \rho, \gamma)$, where $\mathcal{S}$ is the set of states, $\mathcal{A}$ the set of actions, $\mathcal{T}(s|s, a)$ the transition function, $r$ the reward function, $\rho(\cdot)$ the initial state distribution, and $\gamma$ the discount factor. For the problem setting as shown in Fig. 1b, we need to consider two MDPs. Assuming the two MDPs share the same $\mathcal{S}$, $\mathcal{A}$, $\rho$, and $\gamma$, one is the agent's model $\mathcal{M}_A$ with the true domain dynamics $\mathcal{T}_A$ and the engineered reward function $r_A$, and the other is the human's model of expectation $\mathcal{M}_A^H$ with her belief about the domain dynamics $\mathcal{T}_A^H$ and the human's reward function $r_H$. For explicable policy search, $\mathcal{T}_A$, $\mathcal{T}_A^H$, and $r_H$ are *unknown*. Note that our method can be easily applied to a model-based learning setting when $\mathcal{T}_A$ is known.

**Definition 1. Explicable Policy Search (EPS)**  *is the problem of searching for a policy via learning to maximize two objectives: the expected cumulative reward, and a **policy explicability score** between the agent's policy under $\mathcal{M}_A$ and the expected policy under the human's model $\mathcal{M}_A^H$.*

In this paper, we consider a linearly weighted sum of the two objectives:

$$\pi^* = \arg\max_{\pi} \mathbb{E}_{\pi, \mathcal{T}_A} \underbrace{\left[ \sum_t \gamma^t r_A(s_t, a_t) \right]}_{cumulative\ reward} + \lambda \underbrace{\mathcal{E}(\pi, \mathcal{M}_A, \pi^H, \mathcal{M}_A^H)}_{policy\ explicability\ score}, \tag{1}$$

where $\pi$ and $\pi^H$ denote the agent's policy and human's expected policy, respectively. We combine the two objectives linearly via a *reconciliation factor* $\lambda$ to be consistent with the literature [48, 25] and simplify the technical development.

Next, we motivate the design of our explicability score. Intuitively, a data generation model can better explain the data when it assigns the data with a high probability. Similarly, under the assumption that the human generates her expectations based on $\mathcal{M}_A^H$, a behavior (i.e., a trajectory) $\tau$ is likely to be generated by the human's model (or expected by the human) when it is associated with a high probability in the model. In contrast, when the trajectory has a low probability, we could say that it is "inexplicable" with respect to the model. As a result, the probability of the trajectory in the human's model ($\mathcal{M}_A^H$), denoted by $p_A^H(\tau)$, may be considered as our objective at first thought and its expectation over the distribution of trajectories as the explicability score (i.e., $\mathbb{E}_{\pi, \mathcal{T}_A} p_A^H(\tau)$). However, using $p_A^H(\tau)$ only ignores an important aspect of explicability under the model reconciliation setting considered since it involves two models, which requires it to be *contrastive*. More intuitively, for a policy to be the most explicable, a trajectory with a given probability in the human's model $\mathcal{M}_A^H$ should also appear with the same probability in $\mathcal{M}_A$. Appearing more or less likely in $\mathcal{M}_A$ should result a reduced explicability score. Hence, evaluating the explicability of a policy under such a requirement becomes evaluating the closeness of the two distributions. This observation naturally equates the choice for our explicability score to the negative KL-divergence of the two distributions of trajectories under the agent and human's models, respectively.

Hence, we introduce the *policy explicability score* for stochastic environments, which differs fundamentally from the explicability scores defined in the prior work for deterministic domains. In [48, 25], the explicability score has been considered as a similarity metric between the agent's plan and the expected plan. Policy explicability score extends such a metric to consider plan distributions. They are *orthogonal* aspects of explicability that are equally important. Using the newly established policy explicability score, an agent that learns to maximize it would learn to *replicate* (as closely as possible) what the human expects the agent to behave. When the two distributions match exactly, the policy explicability score would be 0, exerting no influence on the final policy as desired. Eq. (1) becomes:

$$\pi^* = \arg\max_{\pi} \mathbb{E}_{\pi, \mathcal{T}_A} \left[ \sum_t \gamma^t r_A(s_t, a_t) \right] + \lambda \cdot -\mathcal{D}_{\mathrm{KL}}(p_A(\tau) \| p_A^H(\tau)), \tag{2}$$

where $p_A(\tau)$ and $p_A^H(\tau)$ denote the distributions of trajectories under $\mathcal{M}_A$ and $\mathcal{M}_A^H$, respectively.

*Remark*: Depending on whether the agent's state and action are observable, the distribution of trajectories may be computed differently. When both the state and action are observable, the human would be able to contrast both the agent's action and resulting state with her expectation; otherwise, only the observable part needs to be considered. In the following discussion, we assume that both the state and action are observable. For explicable policy search, the implication here is that the agent optimizing Eq. (2) has the incentive to avoid parts of the state space where either its policy differs from the human's expectation or the domain dynamics models differ.

## 4 Explicable Policy Search (EPS)

To present our learning method, we start by expressing the probability distribution of the agent's trajectories and human's expectation. We parameterize the agent's policy using $\theta$. The probability of realizing the agent's trajectory $\tau$ with $\pi_\theta$ (and similarly for the human's expectation) is:

$$p_A(\tau) = \rho(s_0)\prod_t \mathcal{T}_A(s_{t+1}|s_t,a_t)\pi_\theta(a_t|s_t), \quad p_A^H(\tau) = \rho(s_0)\prod_t \mathcal{T}_A^H(s_{t+1}|s_t,a_t)\pi_A^H(a_t|s_t). \quad (3)$$

Given these two distributions, we can now derive our solution for EPS. Since both objectives in Eq. (2) are expressed as expectations over the same distribution of trajectories (i.e., $p_A(\tau)$), we can combine them after rewriting the policy explicability score in Eq. (2) as follows:

$$-\mathcal{D}_{\mathrm{KL}}(p_A(\tau)\|p_A^H(\tau))$$
$$= -\mathbb{E}_{p_A}\left[\sum_t \log \mathcal{T}_A(s_{t+1}|s_t,a_t) + \log \pi_\theta(a_t|s_t) - \log \mathcal{T}_A^H(s_{t+1}|s_t,a_t) - \log \pi_A^H(a_t|s_t)\right] + C,$$
$$(4)$$

where $C$ is a constant. The main challenge in solving the optimization problem in Eq. (2) now lies in the fact that neither the human's policy $\pi_H$ nor her belief about the domain dynamics $\mathcal{T}_A^H$ are known.

### 4.1 Surrogate Reward Function

Eq. (4) can be merged into Eq. (2) such that the log terms essentially reshape the engineered reward function. In such a case, a straightforward approach is to learn the human's belief about the domain dynamics and her expected policy separately based on human feedback, while maintaining estimates of the true domain dynamics and the current agent's policy. While possible, it is inefficient and unnecessary. Instead, we propose to use a surrogate reward function. The goal is to learn such a function $u_H$ that retains the necessary and sufficient information about the human's belief and the expected policy for policy search, i.e., $u_H \doteq \log \mathcal{T}_A^H(s_{t+1}|s_t,a_t) + \log \pi_A^H(a_t|s_t)$. Next, we reveal how such a goal can be achieved by manipulating a preference-based formulation.

Intuitively, we learn a reward function that alone can explain the human's expectation of the trajectories–a reward function that introduces the same distribution of the expected trajectories. At the same time, we must take care to constrain the complexity of human feedback data to make it practical. A popular approach based on simple human feedback is preference-based RL [46, 10]. For EPS, instead of soliciting human preferred trajectories to learn the human's reward model, we can present pairs of trajectories and ask humans to comment on which one is *more expected*. Then, we can fit a reward function that prefers generating more expected trajectories. More specifically, we correlate the distribution of the expected trajectories with the surrogate reward function as follows:

$$p_A^H(\tau) \propto \exp\left(\sum_t u_H(s_t,a_t)\right). \quad (5)$$

**Proposition 1.** *There exists a **unique** reward function $u_H$ such that the trajectory distribution under the softmax human preference model with $u_H$ described above matches with the human's expected trajectory distribution given in Eq. (3).*

When we compare mathematically the two equations of $p_A^H(\tau)$ (i.e., Eqs. (3) and (5)), we see that the only way to match the distributions is by satisfying:

$$\sum_t u_H(s_t,a_t) = \sum_t \log \mathcal{T}_A^H(s_{t+1}|s_t,a_t) + \sum_t \log \pi_A^H(a_t|s_t) + C_1.$$

One straightforward way for this is to set $u_H(s_t, a_t) = \log \mathcal{T}_A^H(s_{t+1}|s_t, a_t) + \log \pi_A^H(a_t|s_t)$. In the other direction, since the trajectories may be of various lengths in general (including the length of 1 as special cases), we can conclude further that:

$$u_H(s_t, a_t) = \log \mathcal{T}_A^H(s_{t+1}|s_t, a_t) + \log \pi_A^H(a_t|s_t) + C_1.$$

Plugging this into any trajectory of length greater than 1, we can conclude that $C_1 = 0$. To learn $u_H$, we learn to maximize the likelihood of the human data. This is similar to MaxEnt IRL [49] such that the reward function is learned to exhibit no preferences beyond matching the human's feedback data.

This means that $u_H$ in such a case becomes equivalent to $\log \mathcal{T}_A^H$ and $\log \pi_H$ for representing the distribution of expected trajectories, which implies that the information needed from $\mathcal{T}_A^H$ and $\pi_H$ for optimizing Eq. (2) can be substituted with $u_H$. To learn the unique surrogate reward function and avoid the non-identifiabillity issue [40], potentially, requires the reward function to be globally optimized over all trajectory pairs, which is not feasible. In our implementation, we simply normalize the learned rewards. Now, we rewrite Eq. (2) based on the above result:

$$\theta^* \doteq \arg\max_{\pi_\theta} \mathbb{E}_{p_A}\left[\sum_t \gamma^t\Big(r_A(s_t, a_t) + \lambda\big(u_H(s_t, a_t) + \mathcal{H}_{\mathcal{T}_A}[s_{t+1}|s_t, a_t] + \mathcal{H}_{\pi_\theta}[a_t|s_t]\big)\Big)\right]. \quad (6)$$

Note that Eq. (6) is an approximation of the original objective in Eq. (2) by applying discounting to the surrogate reward function and entropy terms. This allows us to use them to reshape the reward function. We can view this objective function as two parts. The first two terms, $r_A(s_t, a_t) + \lambda u_H(s_t, a_t)$, requires the agent to maximize rewards from two sources: the engineered reward and the surrogate reward learned from human feedback. These two reward functions are weighted according to the reconciliation parameter as expected. The second part of this objective function is $\mathcal{H}_{\mathcal{T}_A}[s_{t+1}|s_t, a_t] + \mathcal{H}_{\pi_\theta}[a_t|s_t]$, which are two entropy terms. The first term is for the agent's domain dynamics and the second term is for the target policy.

The second entropy term for the target policy (referred to as "*policy entropy*") is well studied in the maximum entropy (MaxEnt) RL framework. [20, 21], However, we note that this term in our work is derived for a completely different reason. In MaxEnt RL, maximizing $\mathcal{H}_{\pi_\theta}$ encourages the agent to explore during learning and increase robustness. Given a fixed surrogate reward function, this term in EPS likewise encourages stochasticity in the agent's policy to increase robustness. A more distinguishing feature of explicable policy search lies in the first term. Maximizing $\mathcal{H}_{\mathcal{T}_A}$ encourages the agent to prefer parts of the environment where there is more stochasticity, which we refer to as "*environment entropy*". The incentive here is to reduce the influence on the policy explicability score due to differences in the agent's policy and the expected policy. Intuitively, at more stochastic parts of the environment where action choices for the agent matter less, the agent's trajectory is more likely to match with the human's expectation even when the agent's policy and the expected policy differ.

**Expectation-based Preferences:** For learning $u_H$, we apply a preference-based learning framework. We request humans to provide their feedback on which segment in a pair of segments $\{(\sigma^1, \sigma^2)\}$ extracted from the agent's trajectories *is more expected*. To consider noise in human feedback, the human's expectation preference is formulated as follows, similar to that in [10, 31, 3, 4]:

$$\hat{P}[\sigma^1 \succ \sigma^2] = \frac{\exp \sum u_H(s_t^1, a_t^1)}{\exp \sum u_H(s_t^1, a_t^1) + \exp \sum u_H(s_t^2, a_t^2)}$$

We learn $u_H$ to minimize the cross entropy between our prediction of the expectation preferences and feedback data. In order to efficiently solicit the human's expectation preference for the agent's behavior, we leverage uncertainty-based sampling [10, 28] to select traces for the human in an active learning fashion. To estimate $\mathcal{T}_A$, we assume it follows a Gaussian distributions and model it using a two-headed neural network that outputs its mean and logarithm of standard deviation given the state and action. The agent interacts with the environment and collects transition data for learning. $\mathcal{T}_A$ is estimated using a data-driven method by minimizing the $L_2$ one-step prediction loss [26]. We use Soft Actor Critic (SAC) [21] for policy learning but other policy search methods are also applicable. To alleviate the issue of non-stationary prediction of expectation preferences during learning, we relabel the data samples every time $u_H$ is updated [27]. The algorithm for EPS is in the appendix.

# 5 Evaluation

We evaluate our method on a set of continuous navigation domains with synthetic human models and a simulated autonomous driving domain with a human subject study. The study is IRB approved and all protocols have been followed. The synthetic experiments are used to validate the effectiveness of our method for searching for explicable policies. The user study is to 1) confirm that our beliefs about domain dynamics can be inaccurate or biased, which can affect our judgements and lead to severe consequences if ignored, and 2) show that our method can effectively address such hidden issues by achieving an intelligent reconciliation between the task performance and human's expectation.

## 5.1 Synthetic Navigation Domains

We conduct experiments on four navigation domains with continuous state and action spaces, as illustrated in Fig. 2. More detailed descriptions of our domains are in the appendix. For all the domains, the agent starts from the upper left corner and aims to navigate to the goal area (depicted in green). We introduce various human inaccurate beliefs about domain dynamics for these four domains[1]. For Domain 1 (D1) and Domain 2 (D2) (i.e., the first two rows in Fig. 2), the human believes that the agent is more likely to fall into the pit when it is close by; the truth is everywhere is the same. For Domain 3 (D3) and Domain 4 (D4) (i.e., the third and fourth row in Fig. 2), the human's belief is that the agent could easily slip on ice. Moreover, the human believes that the agent can readily handle sandy roads. The truth is that the agent is proficient on ice but can easily get stuck on sand in D3. For D4, the agent is in addition capable of handling sand albeit being more costly. The human's expectation preferences for the agent's behaviors is generated synthetically with respect to the human's model.

### 5.1.1 Baselines

We compared our method to three baselines to illustrate its advantages compared to the traditional RL methods that do not consider model differences: Soft Actor Critic (SAC) [21], Deep RL from Human Preferences (DRLHP) [10], and Policy Shaping (PS) [18]. SAC optimizes policy with respect to the engineered reward function and true domain dynamics without considering the human's expectation. DRLHP uses human expectation-based preferences for the agent's behavior to estimate a reward function and applies it to guide policy search. It ignores the engineered reward function. For SAC and DRLHP, we could not use reward signals from both sources due to their different formats. The closest competitor to EPS is policy shaping. Policy shaping learns two policies using reward signals from the environment and human expectation preferences, respectively. A combination policy is then obtained by multiplying them together. It also seeks to combine different information sources that are however assumed to be consistent. That is why it has difficulty handling situations where the human's expectation and agent's optimal behavior conflict (see D3 in Fig. 2).

### 5.1.2 Results and Discussion

Table 1 shows the average return and policy explicability score of EPS compared to the baselines over 100 rollouts, with the sample standard deviations. The policy explicability score is computed using Eq. (4) based on the synthetic human models. For EPS, the reconciliation parameter $\lambda$ is tuned manually to show the different behaviors compared to the baselines in Fig. 2, with $\lambda \in [2.0, 2.8]^2$. For all domains, EPS achieved the highest policy explicability score and followed the best performer for average re-

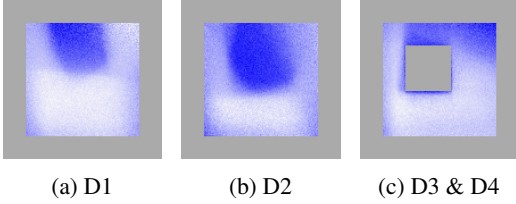

(a) D1      (b) D2      (c) D3 & D4

Figure 3: Visualization of the learned surrogate reward functions for D1-D4. The darker the lower reward value and the brighter the higher.

turn. Note that SAC also achieved good explicability scores in the domains since the synthetic human models did not penalize the optimal behaviors as much as they could (i.e., the human beliefs were not

---

[1]For simplicity, we modified the human's belief about the domain dynamics only while keeping the reward functions the same for the two models. This should not impact the evaluation since the surrogate reward function can encode information for both for policy search.

[2]The spectrum of policies generated by difference values of $\lambda$ is included in the appendix.

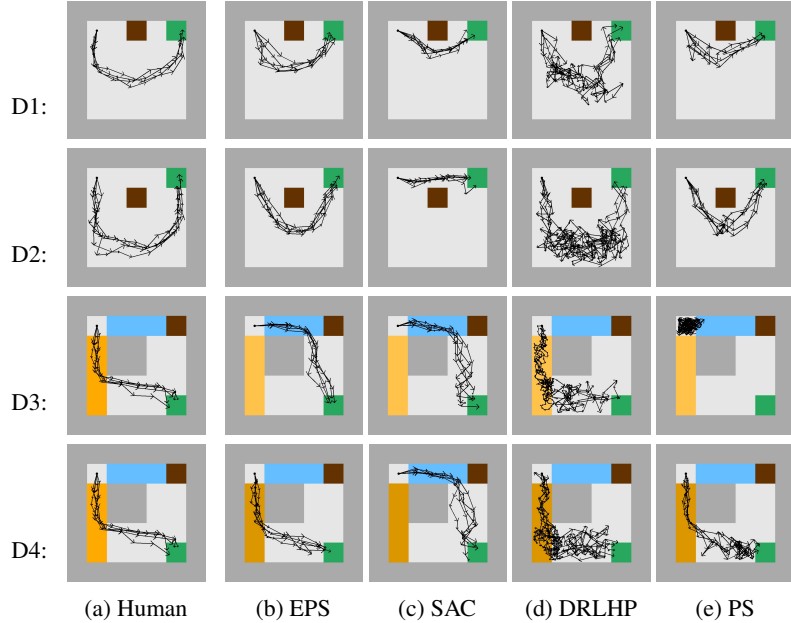

|  | (a) Human | (b) EPS | (c) SAC | (d) DRLHP | (e) PS |

Figure 2: Comparison of different learning methods with human's expectation (left). The dark grey area represents the walls. The brown area is the pit (with -100 penalty) and the green area is the goal (with +100 reward). The blue area represents icy roads and the yellow area represents sandy roads.

| Domain | Avg. Return (Std) | | | | Avg. Explicability (Std) | | | |
|---|---|---|---|---|---|---|---|---|
| | EPS | SAC | DRLHP | PS | EPS | SAC | DRLHP | PS |
| D1 | 95.53 (1.88) | **97.46 (1.53)** | 74.51 (29.92) | 93.91 (3.55) | **94.87 (1.95)** | 92.57 (1.48) | 73.01 (28.75) | 89.17 (4.82) |
| D2 | 94.07 (2.43) | **95.65 (1.98)** | 23.28 (43.71) | 95.55 (3.17) | **92.10 (2.44)** | 91.00 (3.15) | 20.94 (43.97) | 90.66 (3.94) |
| D3 | **96.80 (1.16)** | 94.47 (2.77) | -50.76 (56.49) | -178.76 (63.88) | **94.72 (1.43)** | 91.88 (3.32) | -50.79 (56.44) | -181.46 (63.87) |
| D4 | 92.37 (1.80) | **93.27 (2.03)** | 26.52 (49.02) | 90.11 (14.79) | **92.17 (1.86)** | 90.77 (2.44) | 26.49 (49.03) | 90.11 (16.63) |

Table 1: Comparison of EPS to baselines using averaged return and explicability score.

modeled to be much more stochastic than the ground truth). PS performed well on most tasks while being considerably worse than others in D3. Due to the conflict between the human's expectation and task priorities in D3, the method failed to achieve a meaningful combination. DRLHP performed poorly on all the domains because preference-based learning methods have difficulty dealing with domains with sparse or delayed rewards, which makes credit assignment challenging and results in large variations. Since EPS also applies a preference-based learning method to learn the surrogate reward function, we demonstrate this using the learned function. As shown in Fig. 3, this function has a significant amount of uncertainty so is not ideal for guiding policy search. Although it adds variability into policy learning, it serves very well as an *auxiliary* objective by informing the agent where the human expects it to perform.

We show sampled trajectories for EPS, the baselines, and the human's expectation computed using the synthetic human models in Fig. 2. In D1 and D2, SAC agent always chooses the shortest path while EPS agent takes a detour that bypasses the pit and hence is more explicable. In D3 and D4, SAC agent chooses the path passing through the top since the agent can get stuck on sandy roads in D3 and the top path is more efficient in D4. It is worth noting that EPS agent makes different decisions on these two domains. It selects the top path in D3 when the sandy road is difficult to navigate even though it is against the human's expectation. This is because if the agent conforms to the human's expectation, it would result in a huge loss to the engineered reward. However, when it can better navigate on sand in D4, it chooses to respect the human's expectation to be more explicable, at the cost of lower task performance. Policy shaping agent is also successful in D4, but gets stuck in D3, because simply multiplying two different policies could result in a new policy that is uninformative, irrespective of the $\lambda$ used (e.g., when we have $p_1 = (0.1, 0.9), p_2 = (0.9, 0.1)$, the resulting policy from multiplying them would always be $p = (0.5, 0.5)$), which can lead to poor behaviors. In summary, we show that EPS can successfully generate effective behaviors that achieve an *intelligent* reconciliation between

the human's expectation and task priorities. It can generate novel reconciled behaviors (D1 and D2), stick to the task priorities (D3), or follow the human's expectation (D4), as appropriate.

## 5.2 Autonomous Driving Domain

We used a simulated autonomous driving domain [29] to evaluate our method with human subjects and demonstrate its real-world relevance, as illustrated in Fig. 4. The state space is featured by the position and velocity of the ego-vehicle and nearby vehicles. The action space consists of five discrete actions. Initially, the autonomous driving vehicle (green) is running on the highway with a car in front of it (blue) on the same lane and with the same speed. The task of the autonomous driving agent is to handle situations when the front car slows down quickly and abruptly. One common type of cognitive bias is the *availability bias*, which reflects humans' tendency to overestimate the likelihood of events with greater availability in memory. Such biases can occur in driving since we regularly drive under familiar conditions. To design an experiment where such biases are present, we considered scenarios with a regular driving condition and a sleety condition where the vehicle's domain dynamics became more stochastic due to slippery roads. With the availability bias, however, humans are likely to make the same decisions under these similar but in actuality different conditions, leading to safety risks.

The user study consists of two phases: training and testing. At the beginning of the training phase, we requested the participates to provide the importance ratings for several factors governing the autonomous driving behavior [9] in a 5-point Likert scale, such as average speed (3.42), distance to the front car (4.57), relative speed to neighboring cars (3.28), and lane following (3.78). The average participants' responses are shown in the parentheses above. Their responses served as the engineered reward model ($r_A$) and are linearly combined. For example, the distance to the front car is directly associated with collision and hence given a larger weight. Before human data collection, we also presented the participants with the vehicle's behavior after braking under the regular condition when the front car slows down suddenly. Then, given the information that it is driving on a sleety day, we selected segments that showcased different vehicle behaviors after braking (i.e., braking behaviors under different effectiveness). The participants were asked to select which one matched their expectations the most. Their responses were used to validate the availability biases in this experiment. Then, we collected human data by actively selecting pairs of trajectory segments for the participants to compare with. The collected data was used to train our method and the baselines. Then, we showed rollouts of the learned policy for each method to new participants and asked them to rate those rollouts in the testing phase.

### 5.2.1 Results and Discussion

We published the experiments on Amazon Mechanical Turk (MTurk). To sift out invalid responses, a sanity check demonstration was added that showed a collision, which should not be preferred under any situation. We recruited 15 participants for training, and one failed the sanity check. For the bias validation question, 12 out of 14 valid participants chose that the ego-vehicle would slow down effectively on a sleety day when braking. This reflects the availability bias that the participants had, which can lead to collisions on a slippery road if ignored. We also noticed that the participants preferred to brake than steer in general during training, which accords with the participants' responses that attached a high importance rating to distance to the front car and lane following.

The testing phase occurs on a sleety day. The behavior of EPS, SAC, and DRLHP agents are illustrated in Fig. 4. SAC agent steers immediately when the front car slows down. It is the most efficient behavior with the most return on a slippery road since braking would not be effective. DRLHP agent, on the other hand, chooses to brake while staying in the same lane. Such a behavior, however, is more likely to lead to a collision (risk to human passengers). As we can see from Fig. 4, the agent is getting dangerously close to the vehicle in the front. EPS agent first chooses to brake (to be explicable), and then steers to the other lane (to stay safe and continue moving forward), which maintains both explicability and task efficiency. We demonstrated these rollouts in the testing phase to new participants. We informed the participants that the vehicle was running on a sleety day. Each participant was required to provide ratings to all the demonstrations ranging from 1 to 10. We obtained 11 valid responses. Interestingly, the participants rated EPS agent the most preferable, followed by DRLHP and SAC agents. The average rating for each agent and its standard deviation are shown in Fig. 4. We note that the results contradict with the importance ratings solicited for the engineered reward model, which should have led to preferring the steering behavior as chosen by the

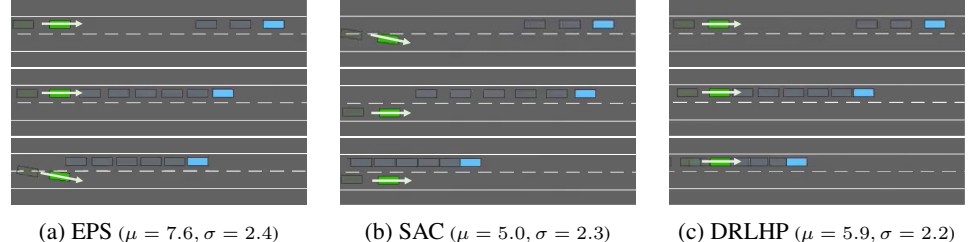

(a) EPS ($\mu = 7.6$, $\sigma = 2.4$)   (b) SAC ($\mu = 5.0$, $\sigma = 2.3$)   (c) DRLHP ($\mu = 5.9$, $\sigma = 2.2$)

Figure 4: Agents' behaviors in the autonomous driving domain illustrated in three characteristic steps from the top to bottom for each agent. The agent's name is followed by its average rating and standard deviation. We set the significance level at $\alpha = 0.05$ for all statistical tests. We performed the one-way ANOVA test on the data. The result [$F(2, 30) = 3.665$, $p = 0.037$] shows a significant difference among the three groups. We conducted the Shapiro-Wilk and Levene's tests for checking the normality and homoscedasticity assumptions, respectively. The results show that our data satisfies the normality [$W(33) = 0.948$, $p = 0.119$] and homoscedasticity [$F(2, 30) = 0.168$, $p = 0.846$] assumptions of one-way ANOVA [42].

SAC agent on slippery roads. This further suggests that the participants had a biased belief about the vehicle's dynamics (i.e., braking is as efficient under the regular condition as under the sleety condition). Regardless, our agent chose the behavior that was both explicable and safe.

## 6 Discussions and Conclusions

In this paper, we introduced and formulated the problem of explicable policy search that considers model differences between the learning agent and its human observer. We developed an efficient solution by learning a surrogate reward function that was then used to recover an explicable policy. Our method significantly extends explicable planning to stochastic environments in an RL setting with continuous state and action spaces. We evaluated in simulations and with human subjects. Results showed that our approach could better handle situations under model differences than several baselines and thus contributed a critical tool to achieving explainable human-agent interaction.

**Limitations and Future Work.** The assumptions underlying our work include that the human generates expectation based on her objective and belief about the domain dynamics in a noisily rational way, and the human has a softmax preference model. While such assumptions were commonly made in prior work, it would be useful to consider situations where such assumptions do not hold, e.g., when the human is biased in the computational model. Furthermore, we assume that the human's inaccurate belief about the domain dynamics does not easily update as observations of the agent are made. For scenarios where inaccuracies stem from intrinsic cognitive biases that are difficult to change, this is a reasonable assumption. In other cases where discrepancies were due to, e.g., information asymmetries, the belief can change dynamically and needs to be actively monitored [22]. Other possible directions include more complex (such as state dependent) reconciliation of the expected return and policy explicability score, as well as weighted policy explicability scores to incorporate trajectory similarity as considered in prior work [48, 25].

Our approach covers many real-world applications since the human understanding of the agent's model is inevitably different from the ground truth in real-world scenarios. As a result, EPS may sometimes cause the user preferences to be ignored, which may be "unexplainable" but is a necessary cost to pay in many scenarios. In particular, EPS combines the surrogate reward with an engineered reward from the agent's designer to maintain critical features (e.g., safety) while respecting user preferences whenever feasible. Note that the combination of such two reward functions should not be replaced by the user's reward even if it can be perfectly estimated due to complexities such as human biases and misunderstandings of the domain. However, real-world deployment of EPS would require more work, such as an interactive design that explains to the user why her preferences are ignored.

**Societal Impacts.** Our research is fundamental in nature without notable negative societal impacts. However, it is conceivable that our approach may be used to misguide RL agents to ignore human preferences in adversarial settings. Further investigation is needed to understand such a possibility.

**Acknowledgement.** This research is supported in part by the NSF grants 1844524, 2047186, the NASA grant NNX17AD06G, and the AFOSR grant FA9550-18-1-0067. The authors would also like to thank the anonymous reviewers for their helpful comments and suggestions.

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
