# Explicable Policy Search

**Ze Gong**
Arizona State University
Tempe, AZ 85281
zgong11@asu.edu

**Yu Zhang**
Arizona State University
Tempe, AZ 85281
yzhan442@asu.edu

## A    Appendix

In this appendix, we include details of experiments and user study discussed in the paper, and present additional experimental results.

### A.1    Synthetic Experiments

To validate our method, we created a set of continuous navigation tasks with synthetic human models. The environment of all the task domains is in the form of a $7 \times 7$ continuous grid-world. The state space consists of the 2D location of the agent in the domain. The action space consists of velocity and angle of the agent's move. Moreover, we added Gaussian noises to each move to simulate stochasticity in the real-world. We conducted experiments on four navigation domains with continuous state and action spaces:

- **Domain 1 (D1):** This domain is adapted from the classical cliff walking domain [10]. There is one pit area with $-100$ penalty and a goal area with $+100$ reward in the environment. The reward of each $(s, a)$ pair depends on how much the action $a$ forwards the agent towards the goal when it is at state $s$. Moreover, there is an additional living reward (i.e., $-1$) for each step. The environment is shown in Figure 1a. The dark grey area around represents the walls. The brown block is the pit and the green block in the upper right corner is the goal. The agent starts from the upper left corner and aims to navigate to the goal.

- **Domain 2 (D2):** Similar to domain 1, this domain contains one pit area and a goal area. The true rewards and dynamics are the same as domain 1. The only difference is that the location of the pit area is moved one block down. Now, the agent has two possible ways to reach the goal that are separated by the pit. The environment is shown in Figure 1b.

- **Domain 3 (D3):** As shown in Figure 1c, the goal is at the bottom right corner of the environment. There are two paths starting from the upper left corner (i.e., the initial position) to the goal separated by an impassable area in the middle. The path passing through the top is an icy road (depicted in blue) and the pit is at its right end. The other path passing through the left is covered with sands (shown in yellow). The environment reward is the same as the first two domains. However, the dynamics model is different while navigating on different road conditions. In this domain, the agent is adept at moving on the icy road while extremely clumsy (i.e., can easily get stuck) on sand.

- **Domain 4 (D4):** The domain is a modified version of domain 3. The only difference is that the agent is now more proficient with sands–it is slower on sand but still maneuverable. The environment is shown in Figure 1d where the sandy road on the left is illustrated in darker yellow to indicate that it is more navigable for the agent.

36th Conference on Neural Information Processing Systems (NeurIPS 2022).

**Algorithm 1** Explicable Policy Search
___
1: **Initialize** the agent's policy $\pi_\theta$, true domain dynamics $\mathcal{T}_A$, surrogate reward function $u_H$, and an empty set $D$.
2: **for** $t = 1 \cdots max$ iterations **do**
3:  Collect samples from environment using $\pi_\theta$ and add them to $D$.
4:  Learn $\mathcal{T}_A$ using $D$.
5:  **repeat**
6:   Select $m$ pairs of trajectory segments from $D$ to solicit expectation preferences.
7:   Learn $u_H$ using human feedback data.
8:  **until** presented a predefined number of sample pairs
9:  **if** reached a predefined batch size **then**
10:   Update $\pi_\theta$ based on the reshaped reward in Eq. (6) in the main paper.
11:  **end if**
12: **end for**
13: **return** $\pi_\theta$
___

We built in various human biased models for these four domains[1]. For D1 and D2, the human believes that the agent is more likely to fall into the pit when it is close by (i.e., more stochastic) and the agent can navigate stably while further way. For D3 and D4, the human believes that the agent would easily slip on ice and it is more likely to fall into the pit while moving close by. Moreover, the human believes that the agent can readily handle sandy roads. In general, D1 and D2 is designed to demonstrate that EPS is able to generate novel behaviors other than the optimal agent's policy and the human's expectation. D3 and D4 examines how the EPS agent determines as it has to choose to follow between optimal agent's policy and the human's expectation.

**Implementation Details.** Our implementation follows the Algorithm 1 which includes the following components:

- Learner for environment dynamics: The dynamics function is formulated as Gaussian distributions that are approximated by a neural network. The state-action pair $(s_t, a_t)$ serves as input. The outputs are the mean and logarithm of standard deviation of the next state $s_{t+1}$. The network has two hidden layers with 256-unit each and ReLU activation. Also, the sampled $s_{t+1}$ is rescaled to fit the domain.

- Learner for surrogate reward function: The surrogate reward function is represented by Gaussian distributions and approximated by a neural network with two hidden layers with 256 units each and ReLU activation. The input is $(s_t, a_t)$ and the output are the mean and logarithm of standard deviation of the reward value clipped within a predefined range. Then it is normalized to facilitate policy search. For reward learning, we fit the human data to the preference model and optimize the cross entropy loss as in [3]:

$$\text{loss}(u_H) = - \sum_{(\sigma^1, \sigma^2, \mu) \in D} \mu(1) \log \hat{P}[\sigma^1 \succ \sigma^2] + \mu(2) \log \hat{P}[\sigma^2 \succ \sigma^1].$$

  where $\mu$ is a distribution over $(\sigma^1, \sigma^2)$ indicating which segment the human prefers. $\hat{P}[\cdot]$ is the expectation-based preference model we introduced in section 4.1 in the main paper.

- Policy search. We use SAC with experience replay [5, 6] for policy learning. Our implementation is built on top of [11] with MIT license. Unless otherwise specified, all hyperparameters are taken from [11].

We compared our method with three baselines: SAC [5, 6], DRLHP [3] and Policy Shaping [4]. SAC is build on top of [11]. DRLHP is build on top of [1] with MIT license. For policy shaping, we learned two policies $\pi_A$ and $\pi_H$ based on two information sources: reward from the environment and the surrogate reward learned from human data. These two policies are learned using SAC and DRLHP, respectively. At each step, the resulting policy is a Bayesian combination of these two policies by multiplying them: $\pi \propto \pi_A \times \pi_H$.

___
[1]For simplicity, we modified the human's belief about the domain dynamics only while keeping the reward function the same. This should not impact the evaluation results since we have shown that the surrogate reward function can encode biases in both.

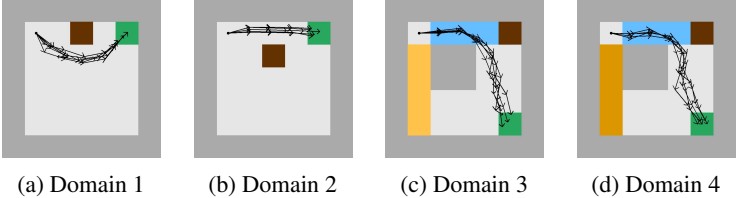

| (a) Domain 1 | (b) Domain 2 | (c) Domain 3 | (d) Domain 4 |

Figure 1: EPS agent's behaviors for different domains when there are no human biases.

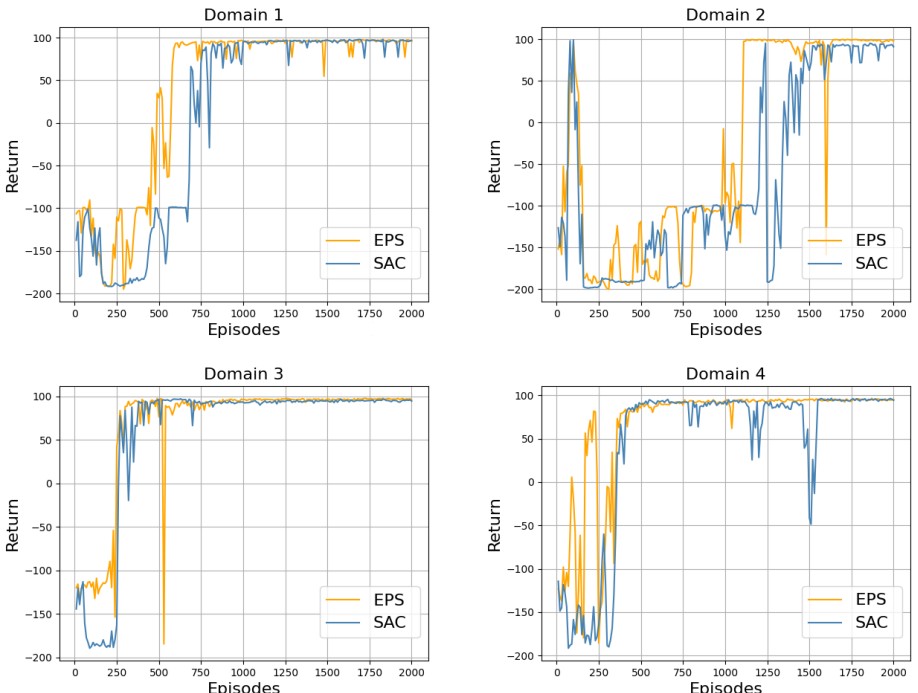

Figure 2: Comparison of the learning process of EPS and SAC in terms of return.

### A.1.1 Synthetic Experiments Without Biases

We show that our method reduces to SAC when there is no human bias. The goal here is to see that our method could recover the agent's optimal policy when human preference data is not biased. Figure 1 illustrates the behaviors of EPS agent without human biases. It successfully recovered the SAC agent's behavior as shown in Figure 2 in the main paper. We also present the accumulative rewards of EPS and SAC during the policy learning process in Figure 2. For all domains, we find that these two approaches are comparable (as expected) with EPS perhaps slightly better. EPS helped the agent learn faster since the surrogate reward function served now as a good inductive bias.

### A.1.2 Sensitivity Analysis of the Reconciliation Hyperparameter

As shown in Eq. (6) in the main paper, the hyperparameter $\lambda$ reconciles between the reward from environment and the surrogate reward learned from the human data. To better understand the effect of this parameter algorithm, we consider a spectrum of the value of $\lambda$ and compare the generated behaviors in all domains. We illustrate the generated behaviors with different $\lambda$ in Figure 3. For all domains, it shows that the lower the $\lambda$, the closer the behavior is to the SAC agent, while the higher the $\lambda$, the closer the behavior is to the human's expectation. The range we tested is $\lambda \in [0.0, 3.0]$. Note that a similar weighting factor may arguably be used in policy shaping to achieve a similar reconciliation effect. However, policy shaping is fundamentally ill-posed (and hence much less robust) when the human's expected behavior and the optimal behavior differ significantly as seen in D3 from Figure 2 in the main paper. In real-world applications, this parameter should be set in a

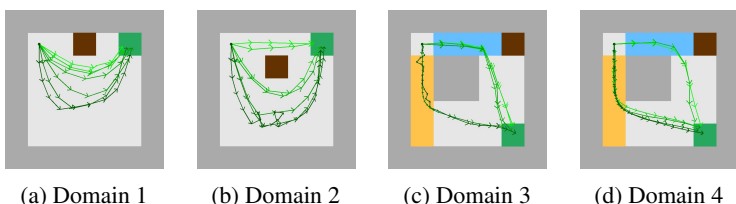

| (a) Domain 1 | (b) Domain 2 | (c) Domain 3 | (d) Domain 4 |

Figure 3: EPS agent's behaviors with different $\lambda$ in all domains. Light green trajectories have smaller $\lambda$ values while dark green trajectories have larger $\lambda$ values.

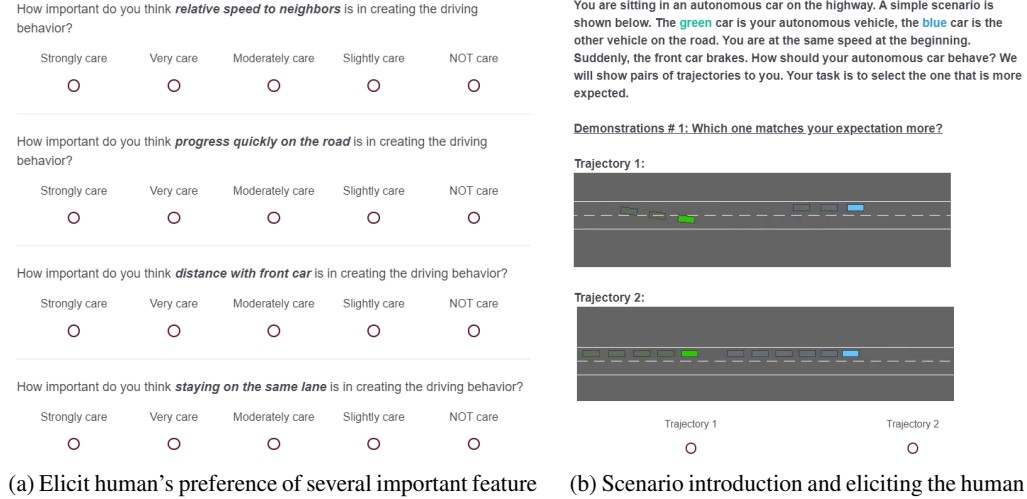

(a) Elicit human's preference of several important feature regarding driving behaviors.

(b) Scenario introduction and eliciting the human expectation-based preference.

Figure 4: Screenshot of the user study on the autonomous driving domain.

domain specific way, similar to the discussion in [12, 7]. In our work, we manually tune the $\lambda$ value for each domain. The automatic tuning of it will be studied for future work.

## A.2 User Study on the Autonomous Driving Domain

Our user study on the autonomous driving domain was meant to evaluate our method with real-world biases. The autonomous driving vehicle is running on the highway with a front car on the same lane. The task of the agent is to handle situations when the front car abruptly and quickly slows down.

We created a new domain which was adapted from the implementation of an autonomous driving environment [8]. The state space contains the position and velocity of the ego-vehicle and nearby vehicles. The action space consists of five actions: accelerate, brake, idle, steer left, and steer right. The dynamics of the vehicles are deterministic and governed by a Kinematic Bicycle Model [9]. We assume the reward function is a linear combination of several features regarding driving behaviors and elicit the human's preference about the importance of these features. Then the true reward function for the vehicle is computed based on the human's feedback that assigned importance ratings to the features.

There were two phases of our user study: training and testing. In the training phases, we elicited the human's feedback on driving behaviors in the regular condition to get the importance ratings and collect the human's preference about the ego-vehicle's behaviors based on a set of trajectory segment pairs. In the testing phase, we demonstrated behaviors of different agents whose policies were learned using EPS, SAC and DRLHP, respectively under the sleety condition (slippery) to the human participants, and asked for their ratings of the behaviors.

Our user study was published on the Amazon Mechanical Turk (MTurk) as shown in Figure 4. We recruited 15 participants for training with an average age of 39. Each of the participant was paid 1 dollar for this 20 minutes task. They were provided with instructions about the scenarios at the

beginning. Next, they were asked to provide their importance ratings for several factors (features) regarding driving behaviors, which are borrowed from [2], in a 5-point Likert scale. These features include average speed, distance to the front car, relative speed to neighboring cars, and lane following. After these questions, we asked the participants to select his or her preferred behavior from a pair of ego-vehicle's trajectory segments. The queries were selected manually. The selected queries covered various scenarios, such as, slowing down by braking, immediately steering to the other lane, and so on.

In the testing phase, we recruited 15 participants with an average age of 38. At the beginning, we informed the participants that the scenarios occurred in the sleety condition. Given the learned policies, we showed the agents' behaviors that were sampled from the policies to participants and asked them to rate the behaviors. The agents' behaviors were illustrated in the main paper. The reconciliation hyperparameter used to generate the explicable policy was 2.0. We presented additional results about the returns besides

Table 1: Comparison between EPS and baselines on an autonomous driving domain.

|  | EPS | SAC | DRLHP |
|---|---|---|---|
| Avg. Rating | 7.6 | 5.0 | 5.9 |
| Avg. Return | 11.0 | 11.9 | 9.6 |

human ratings in Table 1. They showed that the participants rated the EPS agent the most preferable, followed by DRLHP and SAC agents. In terms of the true return, the SAC agent obtained the most return. In addition, DRLHP agent obtained a significantly lower return compared to the other two since it was more likely to collide with the front car when it chose to brake on a slippery road. Our EPS agent appropriately balanced in between the true reward and human's expectation (that are biased).