# OpenReview forum: "Explicable Policy Search"
_NeurIPS.cc/2022/Conference — NeurIPS 2022 Accept_

### Official Review · Reviewer_5GPj · 2022-07-10

**Rating:** 5
**Confidence:** 3
**Soundness:** 2 fair
**Presentation:** 2 fair
**Contribution:** 3 good

**Summary:**

This paper presents a way to make policy search "explicable" to humans, by eliciting their preferences over trajectories and combining a conventional "engineered" reward function with those trajectory preferences. The method is validated in empirical experiments as well as a user study in a small autonomous driving example.


**Questions:**

1) Is the actual contribution here the more efficient way of estimating the human beliefs over dynamics + policy by adding them together into a "surrogate reward function" directly learned by preference learning over trajectories, or the concept itself of using a linear combination learned preferences from [25,7] with a regular (given) RL reward function, or both? Please clarify positioning vs. earlier work in RL.

2) Shouldn't the entropies in Eq. 6 be outside the expectation over p_A? Is this just a typo?

3) Estimating the entropy over the true dynamics seems potentially very difficult in the general case for more complex and higher-dimensional state distributions. It is just briefly mentioned that this is done by learning the true dynamics via a NN with a Gaussian head. For more complex state distributions than narrow trajectory distributions, I wonder how reliable this would be. Any error here seems like it could also affect tuning of lambda.

4) How to set lambda? The first part of the objective is a reward, the second part is a messy approximation to a KL divergence. It is not obvious to me that even if you elicit trajectory preferences from people that making a good trade off here in the end won't still require human input.


**Limitations:**

Sufficiently adequate (pending questions above).

**Strengths And Weaknesses:**

Strengths:
- Appears to be a novel niche in a relevant problem.
- Inclusion of a user study is a nice addition to this kind of paper.

Weaknesses:
- The initial motivation of the approach is weak and confusing. The positioning against related work in RL could also be better. It wasn't clear until the end of the paper that this might be a valid (if rather narrow) problem and that the approach might work.
- The mathematical derivation seems needlessly complex and seems to have a typo or mistake (see below). The technical contributions do not appear very significant.

Details:
- Motivation: The "explicable" solution in the motivating example (Fig 1a). is actually inexplicable to me. Since the proposed approach attempts to plan w.rt. wind, I assume the wind speed is constant (non-constant wind gusts are almost unpredictable and hence unplannable). Having worked with planning for drones, if the wind is constant, but less than the control authority of the drone (here drawn as a quadrotor), it would deviate some depending on the type/parameters of the controller,  strength of wind, and length of the trajectory. However, without knowing those variables, either of the drawn scenarios could happen. In addition, the "expected" dashed line is not straight either but curves in the opposite direction? This example could use a rework as the supposedly explicable and optimal solutions do not appear more explicable or optimal respectively.

- As you appear to use the preference estimation machinery from [25,7], you might want to compare your work against those in the related work. Other RL works have also combined conventional rewards with a KL-divergence objective on trajectories (e.g. variants of Guided Policy Search). Here you are essentially learning this distribution by human preferences learned elicited as in [25,7]?

- The mathematical derivation in 4.1 seems needlessly complicated, you already defined u_H on line 172 to be identical to the log of the human trajectory distribution, so Prop 1 and its proof seems redundant (c.f. e.g. line 172 vs. 188-189).

- Eq. 6 does not appear to be a reformulation of Eq. 2 as claimed. As the authors note they suddenly made the KL-divergence term use the RL discount factor, but more importantly, if you convert the KL-divergence terms from (2) into entropies, shouldn't they be outside the expectation over p_A?

- Eliciting reward functions from human observers appears like a slow process, and the empirical benchmarks are all very small toy examples.

- The autonomous driving user study is a nice inclusion that perhaps should have been given more attention earlier on.

Minor:
- l46: calling it "business" seems informal
- l53: "we assume that the human is nosily rational at generating her expectation of the agent to accommodate her computational limitation" - noisily rational? I am not familiar with this particular model of rationality, it might be good to explain what this means.

---

> ### Author Response · Authors · 2022-08-02
> **Response to Reviewer 5GPj**
>
> Thank you for your thoughtful comments!
>
> For the motivating example, we assume that the drone could be perturbed in the air under normal conditions (i.e.,  without winds), which the human expects. Hence, the expected trajectory is the one with small perturbations (illustrated by the trajectory with small curves). A wind results in higher curvatures in the trajectories. As a motivating example, please note that we did not mean to be precise here and you were right that the feasibility of the trajectories would depend on the many parameters. We will improve the example to make it clear.
>
> 1. The main technical contribution of our work is to introduce the problem of explicable policy search and a feasible framework to solve it. It represents a reconciliation framework that generalizes the conventional preference-based and interactive RL where the effects of the human’s belief about agent’s dynamics have been largely ignored. Ignoring such biases in the belief can lead to disastrous outcomes, as shown in some prior work. The result is an RL agent that must consider reward signals from two sources and balance them intelligently. Thus, our approach opens a valuable direction to generalize RL to apply to human-AI interactive scenarios.
>
> 2. Here we played a small trick. Expectation over p_A of expectation over p_A of the log-likelihood is equivalent to expectation over p_A of the log-likelihood. Thus, we can put the entropy term in the expectation over p_A as well.
>
> 3. This is a valid point and a limitation of our work. Similar to model-based RL methods, when the approximation of the dynamics is poor, it could potentially damage the performance. And this leaves as one of the many research problems to be addressed under EPS, which is however out of the scope here. In this paper, we tested our method in relatively simple domains to demonstrate the importance and relevance of our approach to encouraging research into this problem. We will investigate complex domains and tasks in future work.
>
> 4. Currently, we manually set it to a fixed value. We also tested with a range of different values in the appendix. A more complete treatment is expected in future work. In general, the agent should be able to respect the human expectation of its behaviors on various levels (e.g., conform to the human vs disregard the human’s preference) in different states and under different scenarios. Moreover, the model differences between the human belief and ground truth domain dynamics could be global (i.e., applying to the whole state space) or local. To make the reconciliation factor state and model dependent is an attractive characteristic and potentially a promising research direction.
>
> Line 172 is for identifying the “target form” of the surrogate reward function and we then showed that a preference-based method based on “expectation preference” under certain assumptions would achieve such a goal. Hence, the discussion in 4.1 is necessary but we will improve it to make it clear.
>
> We will add a discussion about noisy rationality. In general, noisy rationality is one type of bounded rationalities due to a limited computational capability, resulting in suboptimal behaviors.

---

> > ### Comment · Reviewer_5GPj · 2022-08-08
> > **Lean accept.**
> >
> > In light of the authors' answer and their answers to the other reviewers, I lean accept instead of reject. I still would have liked to see it evaluated on some more complex real-world problem (e.g. the dynamics/preference modelling might not scale well), but with the added clarifications, the idea seems novel and could be of interest to the community.

---

> > > ### Author Response · Authors · 2022-08-08
> > > **Thanks**
> > >
> > > Thank you for the positive feedback! We agree that how to scale to complex domains and tasks would be quite important and challenging. We will investigate the problem of scalability and evaluate it on more complex real-world problems in future work.

---

### Official Review · Reviewer_TX5X · 2022-07-10

**Rating:** 7
**Confidence:** 5
**Soundness:** 3 good
**Presentation:** 3 good
**Contribution:** 3 good

**Summary:**

This paper develops both a formulation and solution to the problem of what the authors term Explicable Policy Search (EPS), i.e. how to compute policies that maximize a tradeoff between task reward and how expected (or "explicable") a policy is to a human teammate, in order to improve human-robot teaming. The authors formalize this as an objective that combines expected cumulative reward with an "explicability score" for the robot's policy, which is defined as the KL divergence between the robot's policy and the policy that would be expected under the human's model of the environment and the reward function they impute to the robot.

Naively, solving this problem would require access to or inference of the human's model of the environment, and the policy they expect the robot to perform. Instead, the authors show that this latent information can be encoded into an auxiliary reward function, which can be learned from human pairwise comparisons about whether some robot trajectories are more expected than others. Solving the modified MDP with these auxiliary rewards via standard RL methods yields policies that achieve high reward while being "explicable". The authors demonstrate this on a continuous RL test domain using simulated human data where each "human" has a different belief about the domain dynamics, as well as a self-driving car domain with real human feedback. The EPS algorithm outperformed non-"explicable" baselines in terms of the explicability score while still achieving a high amount of cumulative reward, and generated self-driving car trajectories that were more preferred by human raters.

**Questions:**

1. As highlighted above, why call formalize "explicability" as negative KL-divergence, when that more directly captures "expectedness under the human model"? It'd be great if the authors could do more to explain this.

2. Per the discussion above, have the authors considered using negative cross-entropy $E_{p_A(\tau)}[\log p_A^H(\tau)]$ as an explicability score, instead of negative KL-divergence? It's not obvious to me that the incentive to maximize environment entropy or policy entropy (Lines 208-215) is always desired, since this means the agent's behavior will be more unpredicatable!

3. On Line 130, setting the true domain dynamics $\mathcal{T}_A$ to be unknown seems unnecessarily strong a requirement. Surely explicable policy search can be applied in the case where the domain dynamics known as well -- you could just use model-based RL / value iteration instead of model-free methods like SAC.

4. In equation 6, why approximate the original objective by discounting the surrogate reward $u_H$, along with the other entropy terms? It seems like you could still treat them as additional non-discounted rewards, as long as you're optimizing over finite horizons.

5. Also, the phrase "ignoring the influence of the discount factor on the surrogate reward function and entropy term" on Lines 197-198 was confusing to me -- aren't you doing the opposite, by applying discounting to the surrogate reward?

6. Are the standard deviations shown in Table 1 taken over the full set of 100 rollouts (i.e. the sample standard deviation)? Or is that the standard error of the mean? I ask because if it's the latter, then the higher explicability of EPS relative to SAC wouldn't be statistically significant. (No doubt, average-case behavior isn't everything, but is still important).

7. Relatedly, I'm surprised that SAC does so well in terms of explicability score on average. Is there a reason why this is the case? Is it just because in most case, the most optimal behavior happens to be mostly explicable?

**Limitations:**

Authors address a number of limitations which are relevant to extending EPS to real settings where human beliefs about the world may change. Another limitation comes to mind is that EPS is currently restricted to the case where agents have a fixed goal known to the human. How to extend EPS to the case where the agent's goal is not predetermined, and the human may have uncertainty with respect to the agent's goal. Can some combination of explicable and legible planning be used here, and how?

As for potential negative societal impact, the authors claim their work in fundamental, but in my mind it is fairly close to being applied (in e.g. human robot collaboration contexts). I would suggest adding a few words about potential negative impacts from failing to adequately learn and encode the human model through the surrogate reward, or the downsides of trading off too much performance or safety for explicability (which seems to arise in the autonomous car driving scenario).



**Strengths And Weaknesses:**

This paper builds upon prior work on "explicable" and legible planning by introducing a clear and intuitive formulation of "planning in a way that is expected by humans" in the context of stochastic domains where autonomous agents compute policies (distributions over actions given states) and not just plans (ordered sequences of actions). Given that most real-world domains are in fact stochastic, this extension of explicable planning is undoubtedly a useful and significant one.

The mathematical insight used to solve this problem -- showing that human expectations can be encoded as auxiliary rewards -- is also sound, original, clearly explained. The resulting EPS algorithm is well-evaluated on the whole, with clear qualitative demonstrations of how EPS-computed policies lead to more expected behavior given human assumptions about the domain, a user study, and average quantitative performance that is higher in terms of explicability score than a number of baselines (though I have questions about the statistical significance and effect size).

The main concern I have about this paper is surrounding the term "explicable", and how "explicability" is formalized as a (negative) KL-divergence, even though negative KL is better understood as something like "expectedness". I understand that this is similar to how the term "explicable" has been used in prior work in planning, but I initially found it confusing and poorly chosen, because "explicable" evokes the term "explainable", and it's not at all obvious how minimizing KL-divergence (or difference from expected behavior in general) makes behavior more explainable.

Upon further thought however, I do think there *is* an intuitive sense in which the term "explicable" means something like "minimize KL divergence", but it requires a bit of justification / explanation: Typically, when we have a (probabilistic) model of some observations or behavior, we say that the observations are "well-explained" by the model when the observations are assigned high-probability by the model (i.e. their log-likelihood is high). This sort of language is especially apt when the model in question is causal model of the world, and so assigning high probability to the observations corresponds to having found a good causal explanation of the data. In contrast, when the observations have low probability, we might say that they are "inexplicable" with respect to the model.

In the context of EPS, the model in question is the human's intuitive causal model of how the world works $\mathcal{T}_A^H$, and how an approximately rational agent would act in that world to achieve its goals. And so, if a trajectory $\tau$ pursued by an agent has high-probability under the human's model, it is "well-explained" under the model, hence "explicable". Otherwise it is "inexplicable". Hence, the log-likelihood of an agent's trajectory $\tau$ under the human's model $p_A^H(\tau)$ can be treated as an explicability metric.

Now as the authors point out, we don't just want to evaluate specific agent trajectories or plans, but agent policies $\pi_A$, which (when combined with the true environment dynamics) are distributions over trajectories. This motivates taking an expectation over this distribution over how explicable each trajectory is:
  - If the explicability of a single trajectory is defined as $\log p_A^H(\tau)$, then expectation is the negative cross-entropy $E_{p_A(\tau)}[\log p_A^H(\tau)]$ of the human's model over trajectories, relative to the true model.
  - If the explicability of a single trajectory is instead defined as the log odds ratio $\log \frac{p_A^H(\tau)}{p_A(\tau)}$, then the resulting expectation is the negative KL divergence $-D_{KL}(p_A(\tau)||p_A^H(\tau))$.

Either of these gives us a way to evaluate how explicable a policy is relative to the human's model. It's interesting to consider the differences -- and I'd be curious what the authors think about using cross-entropy instead -- but I think the more important point is that this sort of justification for using the term "explicable" is missing from the paper as currently written. I would highly encourage the authors to reframe how their explicability score is introduced and defined, in order to incorporate something like the above. Otherwise, the conceptual gap between "expected by the human" and "explicable" is not an obvious one for readers to bridge. In case helpful, [1] is a paper that connects the quantity of log-likelihood to the concept of "inexplicability", which is formalized alongside other measures of human surprise.

Apart from the above, I only have a number of other minor questions and suggestions, which I'll list in the next section.

[1] Zhi-Xuan, T., Gothoskar, N., Pollok, F., Gutfreund, D., Tenenbaum, J. B. & Mansinghka, V. K.  (2022). [Solving the Baby Intuitions Benchmark with a Hierarchically Bayesian Theory of Mind.](https://social-intelligence-human-ai.github.io/docs/camready_13.pdf) Social Intelligence in Humans and Robots, RSS 2022 Workshop.

---

> ### Author Response · Authors · 2022-08-02
> **Response to Reviewer TX5X**
>
> Thank you for your thoughtful comments!
>
> 1. We will include a more detailed discussion about how the mathematical expression is derived for “policy explicability” as you suggested. Thank you!
>
> Explicability has been used as a metric to quantify the expectedness of the agent’s behavior in the human mind. However, we would like the metric for a single trajectory (whose expectation is computed to form policy explicability) to be specified with respect to both models (i.e., the agent’s model and the human’s model) so that the trajectory can be compared more explicitly. This is because explicability should not only be dependent on how likely a trajectory may appear in the human's model but also on how likely it may appear in the robot's model! For example, in the extreme case when the distributions are the same, we would like it to have no impact on the policy, i.e., policy explicability should be 0. Furthermore, this also opens up the possibility to consider other forms of explicability, such as those involving hierarchies. This is why we chose the KL divergence instead of cross entropy.
>
> 2. Optimizing negative KL divergence would additionally bring about two entropy terms as in Equation 6. Maximizing the environmental entropy encourages the agent to explore the areas that are more stochastic. It offers the agent more freedom to adapt its policy to match the human’s expectations. Maximizing the policy entropy encourages the agent to explore during policy learning to increase robustness While you are right that this may make the agent’s behavior less predictable, it in fact leads to more expected behaviors (i.e., you would expect an agent to slip on icy road even though you may not be able to predict which direction it would slip in)! Investigating the tradeoff between predictability and explicability is another interesting direction.
>
> 3. EPS can be applied with model-based or model-free RL methods. Great comment.
>
> 4. We approximate Eq 6 for problems with infinite horizons. You are right that it does not need to be discounted for finite horizons.
>
> 5. You are right. We will update to resolve the confusion.
>
> 6. The standard deviation is taken over the 100 rollouts. We will make it clear.
>
> 7. The explicability level of optimal behaviors would differ depending on the domain and task. The small difference in explicability score may be due to mixed reasons such as environment setting and the stochasticity in the human’s belief model. In our evaluation, this was due to the fact that the synthetic human models did not penalize the optimal behavior as much as they could (i.e., the human beliefs were not modeled to be much more stochastic than the ground truth). This resulted in the situation that the agent’s optimal behavior may only incur a high cost to explicability in small areas of the environment (e.g., near the cliff).
>
> Considering the extension of explicability to incorporate legibility is an interesting idea. Note that when these are considered orthogonal issues, it is not difficult. However, the challenge here is to model the effect of explicability on the perception/inference of the goal, which could be complicated to model.
>
> We did consider the potential societal impacts. For example, our approach may be used to misguide RL agents to ignore human preferences in adversarial settings but the connection is rather vague so we refrain from discussing it further. We will include more discussion on this and leave further investigations to future work.

---

> > ### Comment · Reviewer_TX5X · 2022-08-08
> > **Response to Authors**
> >
> > Thank you to the authors for the responses to my questions and the careful revisions in response. Note that there is a slight typo in one of the revisions -- I believe the numerator and denominator should be swapped on line 169: $p_A^H(\tau)$ should be the numerator, not the denominator. (I also think the exclamation marks that were added to the the text can be removed.) Apart from that, I believe all my concerns and questions have been adequately addressed by the revisions.

---

> > > ### Author Response · Authors · 2022-08-08
> > > **Thanks**
> > >
> > > Thanks for pointing it out. We revised it in our new version of the manuscript.

---

### Official Review · Reviewer_gaQq · 2022-07-11

**Rating:** 5
**Confidence:** 3
**Soundness:** 3 good
**Presentation:** 3 good
**Contribution:** 2 fair

**Summary:**

The paper proposes an approach for incorporating human preferences into policy search by shaping the 'original' reward function with a 'preference-based' reward function for the human (other than learning the user belief via a respective policy and transition model). The authors learn the human preference-based reward based on available techniques. The approach is evaluated on environments with cont. state and action spaces (navigation & autonomous driving) based on both synthetic human preferences as well as a user study. The results show that the approach works better than pure RL algorithms as well as a preference-based approach.

**Questions:**

* How does the approach related to multi-objective RL? E.g. Yang, R., Sun, X. and Narasimhan, K., 2019. A generalized algorithm for multi-objective reinforcement learning and policy adaptation. Advances in Neural Information Processing Systems, 32.

* How would the system perform against more recent preference-based RL approaches, improving on the used baseline? Could one (with confidence) state that even with close to optimal approximation of the user reward function, the systems would benefit from the combination with the "original" reward?

**Limitations:**

The central limitation which is discussed deals with: The argumentation for learning a user-specific reward function for shaping states that it is unnecessary to learn the user-dependent transition function. Is this always the case? What is the underlying assumption for this? The discussion is surely userful.

It would also be good to discuss the effects of such a straightforward combination of both reward functions. What kind of real world scenarios would this cover? Under which assumptions can a system disregard user preferences (e.g. when safety is incorprated via preferences)?

#### Update after author response & discussion
Again, I thank the reviewers for their answers and changes to the paper. After reading the answers, comments of the other reviewers and especially the changes to the paper, I am wiling to increase my score to above the acceptance threshold. Main concerns have been addressed and approriate changes are in the paper.

**Strengths And Weaknesses:**

Strengths: The topic of the paper is relevant and up-to-date, focussing on including user feedback into RL while taking into accont possible misjudgments of the end-users when giving preferences. The taken approach is sensible, shaping the agent's reward function with the user-dependent reward function. The emprical evaluation comprises a user study, which is very helpful for interactive ML.

Weaknesses: While the evaluation is insightful, it does not compare to more recent (and also already referenced) work on interactive RL, e.g. [24] of the paper. Also, it would significantly help the presentation to include the algorithm (available in the suppl. material) into the main paper. Lastly, I am wondering if such combination of agent- and user- reward function is helpful for the general advancement of RL. It feels like the human should remain in control when configuring the system (which seems to be the poinf of explainable, interpretable, safe RL research), but possibly convinced to change her/his perception. So wouldn't a better way be to incrementally suggest improvements to a policy learned via preference-based RL? Maybe this is equal to the direction mentioned in the future work section (pointing to more complex combinations of both reward functions), but it might also be necessary part to generate value.

Besides, a relevant related work (more into the direction of preference-based RL) is also: Reddy, S., Dragan, A., Levine, S., Legg, S. and Leike, J., 2020, November. Learning human objectives by evaluating hypothetical behavior. In International Conference on Machine Learning (pp. 8020-8029). PMLR.

---

> ### Author Response · Authors · 2022-08-02
> **Response to Reviewer gaQq**
>
> Thank you for your thoughtful comments!
>
> For Questions:
> 1. Our problem is a multi-objective RL (MORL) problem with linearly weighted objectives. There are two objectives for the agent, one is optimality with respect to the engineered reward function, and the other is the explicability of the agent’s behaviors with respect to the human’s expectation. However, the contribution of our work lies in the innovation of the second objective with connection to explainable decision making and the simplification of the optimization problem under hidden objectives. Potentially, insights from MORL may be able to be applied to solve the problem of EPS which is interesting to investigate in future work.
>
> 2. Regarding the “central limitation”: The assumptions underlying our work are that the human generates her expectation based on her objective and belief about the domain dynamics in a noisily rational way, and the human has a softmax preference model. Such an assumption was commonly made in prior work. We will add a discussion about this. Under these assumptions, the influence of the belief about domain dynamics can be “compiled” into a surrogate reward function.
>
> 3. Under our problem setting, the learned surrogate reward function encodes the information of both the human’s “true” reward function and her belief about the agent’s dynamics. When the human’s belief deviates from the ground truth dynamics, the surrogate reward function may fail to capture the human objective using state-of-the-art preference-based RL approaches that overlook the effects of the human’s belief, leading to highly undesirable behaviors. Such an effect is also shown in our evaluation (see Fig. 2).
>
> This approach covers many real-world applications since the human model is inevitably different from the ground truth domain dynamics in real-world scenarios. As you pointed out, EPS may sometimes cause the user preferences to be ignored, which may be undesirable but is a necessary cost to pay in many scenarios. In particular, EPS combines the surrogate reward with an engineered reward from the agent designer to ensure critical features (e.g., safety) while respecting user preference only when possible. Fully addressing such a limitation requires more work, such as an interactive design that explains to the user why her preferences are ignored. Additional discussions on possible extensions to alleviate such an issue are provided below.
>
> Another possible consideration is regarding the reconciliation parameters under the framework of EPS. Currently, we manually set it to be a fixed value, while the agent should be able to respect the human expectation of its behaviors in various levels (e.g., conform to the human vs disregard the human’s preference) in different states and under different scenarios. Moreover, the model differences between the human belief and ground truth domain dynamics could be global (i.e., applying to the whole state space) or local. To make the reconciliation factor state and model dependent is an attractive characteristic and potentially a promising research direction.

---

> > ### Comment · Reviewer_gaQq · 2022-08-10
> > **Good changes & clarifications**
> >
> > I want to thank the authors for their detailed answers and clarifications, as well as the changes in the manuscript. I agree with other reviewers that the changes improve the quality of the submission. Lastly, sorry for the belated reply to the response.
> >
> > My main uncertainty about the paper dealt with seeing the difference between incorporating the "explainable", personalized term and a multi-objective setting (hence the question wrt the relationship, as it might have made sense to formalize under existing, established frameworks), as well as the general possibility to remedy the "felt" unalignment of the end-user through the proposed approach. I did not necessarily intended to point to end-users which are "cognitively bounded", but rather challenged the possibility of convincing a regular end-user to approve the combined policy (i.e. "optimal" plus "explainable"). Maybe I misunderstood, but the preference elicitation step only concerns learning the user-specific reward model and not the combined result, right?

---

### Official Review · Reviewer_PBHQ · 2022-07-11

**Rating:** 3
**Confidence:** 3
**Soundness:** 2 fair
**Presentation:** 3 good
**Contribution:** 3 good

**Summary:**

The authors propose Explicable Policy Search (EPS), a method that aligns the behavior of an RL agent with the one expected by a human observer. More precisely, the goal of the method is to learn a policy that reconciles between maximizing the long-term return and minimizing the deviation from the expected behavior. This is achieved by considering a linear combination of two obectives, namely the cumulative reward and a policy explicability score. Since the human's belied about the domain dynamics is hidden to the agent and cannot be queried, the authors demonstrate that it is possible to use a surrogate reward function that encodes the information needed for EPS.


**Questions:**

* The paper mentions that the actions space in the authonomous driving domain consists of 5 actions, but the actions are only reported in the Appendix. I think it would be helpful to the reader if these actions where listed in the main paper in order to better understand the task.

**Limitations:**

The authors properly described the limitations.

**Strengths And Weaknesses:**

#### Strengths

Overall, the paper describes a nice and interesting work.
I found the paper very well written and easy to follow. The content is presented in a clear and effective way.

The method is sound, simple and elegant.

The paper provides a nice formalism to extend explicable planning to a RL setting with continuos state and action spaces.


#### Weaknesses


The paper is interesting but in the current version the experimental evaluation is a bit lacking. Experiments are mainly performed on synthetic tasks, which are good for a preliminary assessment of the soundness and feasibility of the method, but should be complemented with more analyses. The study with human subjects is interesting but it is only performed on one toy task.

Moreover, the evaluation assesses the deviation from the human's expected behavior in Table 1 based on the explicability score. However, it looks obvious that EPS achieves a better explicability score compared to the other approaches as it is designed to optimize the explicability score. I understand that it is interesting that the averaged return is still high compared to the baselines, but it would be more interesting to perform a human evaluation of the explicability of the approaches.

Overall, I think the paper would benefit from more experiments on real-world data, but even more ablation studies would be interesting. For instance, the appendix includes an experiment on the choice of $\lambda$ for the synthetic domains. I think these experiment could be in the main paper and it would be nice if the paper could include a similar experiment for the autonomous driving task too. Also, more experiments aimed at evaluating the role of the environment entropy (as described in lines 210-215) would be interesting.

---

> ### Author Response · Authors · 2022-08-02
> **Response to Reviewer PBHQ**
>
> Thank you for your thoughtful comments!
>
> First, we would like to stress that our main contribution is to formulate the EPS problem, provide a feasible first solution, and demonstrate its benefits. The aim is to encourage research into this problem, which would be very beneficial to human-AI interaction.
>
> While it is always helpful to include more evaluations and evaluations in more complex domains, the set of results and complexity of the domains are comparable to that of many NeurIPS papers that we have read. If there is any aspect missing that significantly impacted the evaluation of our approach, we would appreciate it if the reviewer can more explicitly point it out.
>
> Including additional human evaluations would be helpful but very costly at the same time. So, it is a common practice to evaluate in relatively simple domains. We would like to point out however that the driving domain was carefully picked to demonstrate the potential of the proposed approach in various domains.
>
> As for the suggestion on ablation studies, both the preference-based and policy shaping methods can be considered as ablated methods, where one uses only one source of information and the other uses a simple method for achieving “reconciliation”.

---

### Official Review · Reviewer_VqJ9 · 2022-07-12

**Rating:** 6
**Confidence:** 4
**Soundness:** 2 fair
**Presentation:** 3 good
**Contribution:** 3 good

**Summary:**

This paper poses a problem for “explicable policy search,” which is defined as the problem of developing a planner (i.e., via reinforcement learning) that produces behavior that are aligned with the expectations of a human observer. The paper develops an approach that infers the human’s latent objective function through preference-based RL (though a Behavior Cloning objective is described first). The paper evaluates the approach in a computational environment with synthetic humans showing promising results. The paper then conducts a human-subject experiment showing that human’s subjectively rate the proposed approach higher than baselines.

**Questions:**

-Line 14, are the human’s expectations truly “hidden” or just partially observable, latent random variable?
-How is proposition 1 fundamentally different than claims of Maximum Entropy Inverse Reinforcement Learning for learning a unique reward function? How does Proposition 1 eliminate the reward ambiguity problem?
-Could the authors provide more information about why EPS did not conform to the human’s behavior in D3?


**Limitations:**

- The authors explicitly have a section on limitations and future work, which is helpful.
- The paper states that the research is “fundamental in nature without notable negative societal impacts.” I appreciate the response, but the response does not seem to be in-line with the expectations for NeurIPS. At one point, GANs might have seemed to be fundamental research, but Deep Fake videos offer the world a tool for disinformation. I would recommend spending additional time pondering whether explicability in planning – as defined here – has benefits and potential harms to society and whether the benefits outweigh the harms. For the record, this reviewer believes the research is ethical. The reviewer is only asking for a more full attempt to fulfill the expectations of NeurIPS.


**Strengths And Weaknesses:**

trengths:
+ The paper’s writing is generally clear with some grammatical issues. The figures were helpful.
+ The topic is timely, as researchers express increasing interest in explainable artificial intelligence in general. Policy explicability is an under-studied topic.
+ The problem formulation is clear, and the approach is well-reasoned.
+ The computational results (Table 1) show competitive returns and superior explicability scores.
+ The visualization in Figure 2 helps provide insight into the behavior of the planners.
+ The paper conducts a human-subject experiment to evaluate the approach, showing that it outperforms baselines with statistically significant results. The study was IRB-approved.

Weaknesses:
-Line 85 states that the surrogate reward function encodes the “necessary information.” Is that true? Is the information learned truly “necessary” or are components of the surrogate reward only approximately correct, superfluous, and/or incongruent with the truly-necessary information? What would the sufficient information be?
-In this paper, the term “explicable” in regards to planning is defined as a plan that aligns with the human’s expectations for what a plan should look like. This is but one form of explicability, or, colloquially, explainability. For example, the paper seems to miss prior work on value-aligned reinforcement learning, such as:

Nahian, M.S.A., Frazier, S., Harrison, B. and Riedl, M., 2021. Training value-aligned reinforcement learning agents using a normative prior. arXiv preprint arXiv:2104.09469.

Brown, D.S., Schneider, J., Dragan, A. and Niekum, S., 2021, July. Value alignment verification. In International Conference on Machine Learning (pp. 1105-1115). PMLR.

The value-aligned RL work seems to be highly similar in idea: creating RL agents that align with human expectations. The paper is also missing prior work on interpretable reinforcement learning, such as:

Silva, A., Gombolay, M., Killian, T., Jimenez, I. and Son, S.H., 2020, June. Optimization methods for interpretable differentiable decision trees applied to reinforcement learning. In International Conference on Artificial Intelligence and Statistics (pp. 1855-1865). PMLR.

Paleja, R., Niu, Y., Silva, A., Ritchie, C., Choi, S. and Gombolay, M., 2022. Learning interpretable, high-performing policies for continuous control problems. arXiv preprint arXiv:2202.02352.

While these other approaches are not “explicable” in the sense that they do not align with a human’s expectations, they are interpretable, i.e. directly “explainable.”

It would have been helpful to include the Brown et al. (2021) method as a baseline and argue why explicable behavior is better than interpretable behavior (relatively to Silva et al., 2020), considering the position of Rudin (2019).

Rudin, C., 2019. Stop explaining black box machine learning models for high stakes decisions and use interpretable models instead. Nature Machine Intelligence, 1(5), pp.206-215.

Rudin, C., Chen, C., Chen, Z., Huang, H., Semenova, L. and Zhong, C., 2022. Interpretable machine learning: Fundamental principles and 10 grand challenges. Statistics Surveys, 16, pp.1-85.

-The approach described in Equation 1 seems quite similar to the mechanisms explored in InfoGAIL (Li et al., 2017). In their paper, Equation 4 shows a discriminator loss for a generative-adversarial approach to essentially Behavior Cloning/Dagger, an entropy bonus, a mutual information term, and the reward function. If we throw out the entropy bonus and mutual information term (which is used for aligning with heterogeneous humans), then we basically recover what we have in Equation 2 of this paper: one term for the expected return (or reward function) and one for aligning with human behavior (behavior cloning).

Li, Y., Song, J. and Ermon, S., 2017. Infogail: Interpretable imitation learning from visual demonstrations. Advances in Neural Information Processing Systems, 30.

- Following up on the previous point, it is fair to note that replacing the divergence term in Equation 2 and arriving at Equation 6 is literally a different formulation (e.g., replacing imitation learning with a preference-based RL, exponential distribution-type reward function inference setup) than in InfoGAIL. However, the improvement seems incremental to replace the KL divergence with U_H, and it would have been helpful to acknowledge this connection.

- Further, the inclusion of entropy (as described in Lines 205-207 further recreates the setup from InfoGAIL without attribution.

- The equation in referenced from [7] goes back at least a decade further to Lucas et al. (2008) and was leveraged again by Brown et al. (2019 and 2020) for preference-aligned RL.

Lucas, C., Griffiths, T., Xu, F. and Fawcett, C., 2008. A rational model of preference learning and choice prediction by children. Advances in neural information processing systems, 21.

Brown, D., Goo, W., Nagarajan, P. and Niekum, S., 2019, May. Extrapolating beyond suboptimal demonstrations via inverse reinforcement learning from observations. In International conference on machine learning (pp. 783-792). PMLR.

Brown, D.S., Goo, W. and Niekum, S., 2020, May. Better-than-demonstrator imitation learning via automatically-ranked demonstrations. In Conference on robot learning (pp. 330-359). PMLR.

- The design of the Likert scales and analysis could be improved. For example, results of test for the ANOVA assumptions (e.g., normality and homoscedasticity) would be helpful. A useful reference would be Schrum et al. (2020).

Schrum, M.L., Johnson, M., Ghuy, M. and Gombolay, M.C., 2020, March. Four years in review: Statistical practices of likert scales in human-robot interaction studies. In Companion of the 2020 ACM/IEEE International Conference on Human-Robot Interaction (pp. 43-52).

Writing:
-There are two uses of the word “it” in Line 4 that may refer to different subjects. Please try to avoid using pronouns with ambiguous antecedents.
-“behavior, due” does not need a comma
-Line 16, “they” is ambiguous

---

> ### Author Response · Authors · 2022-08-02
> **Response to Reviewer VqJ9**
>
> Thank you for your thoughtful comments!
>
> Under the assumption that the human’s belief about domain dynamics is accurate, aligning values becomes equivalent to aligning human expectations under our problem setting. Our approach, at the same time, considers situations where such a belief may be inaccurate/biased. Since our focus has been on such biased beliefs, we did not include much discussion on value-aligned RL. We will incorporate a discussion on the connections to value-aligned RL.
>
> Your point on the differences between explicable and interpretable behavior is an interesting one. In particular, an interpretable behavior may not be expected behavior. However, an expected behavior must be interpretable. In high-stake domains where “surprises” should be minimized if at all possible, explicable behaviors would be more desirable. We will incorporate a discussion on this.
>
> For your question about the “necessary information”, it is a poor choice of words. The surrogate reward function encodes the “necessary and sufficient” information for explicable policy search. Note however that this is the case only under the assumptions that the human’s expectation is generated based on the human’s objective and belief about the agent’s domain dynamics in a noisily rational way, the softmax preference model, and the chosen policy explicability metric.
>
> Thank you for the pointer to Infogail, which we were not familiar with. Infogail uses latent code to discover salient semantic features to accommodate demonstrations from different experts. Even though the aim is quite different from ours, it does seem that there could be interesting connections when considering the biases in the domain dynamics as captured by the latent code. From this perspective, Infogail is focused on “discovering” the differences while we focus on “reconciling” the differences. For example, it seems that Infogail likely would fail in situations where all experts are biased. We will incorporate a discussion on this. We will also include references to Lucas et al. and more on preference-aligned RL.
>
> For Questions:
> 1. “Hidden” is used synonymously as “partially observable” in our work. The preference data (i.e., expectation preferences) provides “observations” for estimating human expectations.
>
> 2. Similar to MaxEnt IRL, the preference model used in section 4.1 does not exhibit any additional preferences beyond matching the surrogate reward function. Trajectory segments with equivalent surrogate rewards have equal probabilities, and segments with higher surrogate rewards are exponentially more preferred. From the uniqueness point of view, they are similar. Thank you for pointing it out. We will make the connection clear.
>
> 3. Proposition 1 shows that the surrogate reward function would not suffer from the reward ambiguity problem under the stated assumptions in our problem setting (e.g., softmax preference model and noisy rationality). However, in practice, these assumptions may not hold in general. Hence, we chose a simplified implementation (see the discussion after Proposition 1), which however may be affected by reward ambiguity.
>
> 4. The agent’s behavior is influenced by both the engineered reward function and the surrogate reward function which encodes the human’s objectives and her belief about the agent’s dynamics. In D3, if the agent conforms to the human’s expectation, it would result in a huge loss to the engineered reward. Thus, our agent that intelligently balances between performance loss and explicability gain chose to not conform to the human’s expectation.
>
> We did consider the potential societal impacts. For example, our approach may be used to misguide RL agents to ignore human preferences in adversarial settings but the connection is rather vague so we refrain from discussing it further. We will include more discussion on this and leave further investigations to future work.

---

> > ### Comment · Reviewer_VqJ9 · 2022-08-06
> > **Thanks**
> >
> > I thank the authors for their helpful response.
> >
> > Did the authors want to take the opportunity to revise their manuscript for the reviewers to review? I would appreciate the chance to see how the proposed changes would be incorporated.
> >
> > Also, did the authors have any thoughts on the Likert scale analysis?

---

> > > ### Author Response · Authors · 2022-08-07
> > > **Thanks! Revision submitted.**
> > >
> > > Thank you for your suggestion! We revised our manuscript and submitted it for review.
> > >
> > > For the Likert scale analysis, we include new results by conducting the Shapiro-Wilk test for checking the normality assumption, and Levene’s test for checking the homoscedasticity assumption. The p-value of the Shapiro-Wilk test is 0.1192 (>0.05) and the p-value of Levene’s test is 0.8465 (>0.05). The results show that our tests satisfy the normality and homoscedasticity assumptions of one-way ANOVA.

---

> > > > ### Comment · Reviewer_VqJ9 · 2022-08-08
> > > > **Response**
> > > >
> > > > Thanks for revising. Feel free to omit the (>0.05) and simply say that significance is set at the \alpha = 0.05 level.
> > > >
> > > > Please use APA format for reporting statistics. For example, a t-test would be something like "t(dof) = X.YZ, p = 0.AB." Please do the same for the Shapiro-Wilk and Levene.

---

> > > > > ### Author Response · Authors · 2022-08-08
> > > > > **Thank you for the suggestion**
> > > > >
> > > > > Thank you for the suggestion! The manuscript is updated.

---

### Author Response · Authors · 2022-08-07
**Revision submitted**

Thank you for all the insightful comments. We revised our manuscript and submitted a new version for review.

---

### Meta-Review · Area_Chair_D3jB · 2022-08-25

**Recommendation:** Accept
**Confidence:** Certain

**Metareview:**

I thank the authors for their submission and active participation in the discussions. This paper studies the problem of devising an RL planner that produces bahviour consistent with human observer preferences. Reviewers remarked that the paper studies a timely problem [VqJ9,gaQq,5GPj], containing clear writing [VqJ9,PBHQ] and useful visualizations [VqJ9], and provides insightful human evaluations [VqJ9,gaQq,5GPj], and that the method is sound and elegant [PBHQ,TX5X]. During AC/reviewer discussion, reviewers TX5X, gaQq and VqJ9 agree that the author response has addressed the main concerns. I am slightly discounting the negative score by reviewer PBHQ as I don't find their suggestion to include more experiments and real world data very concrete or actionable. Thus, I overall see support for accepting the paper and am therefore recommending acceptance while encouraging the authors to further improve their paper based on the reviewer feedback.


**Award:**

No

---

### Decision · Program_Chairs · 2022-09-14

Accept